# Changes in the Composition and Diversity of the Intestinal Microbiota Associated with Carbohydrate Consumption in Type 2 Diabetes *Mellitus* Patients

**DOI:** 10.3390/ijms252212359

**Published:** 2024-11-18

**Authors:** Beatriz Elina Martínez-Carrillo, Amapola De Sales-Millán, José Félix Aguirre-Garrido, Roxana Valdés-Ramos, Flor de María Cruz-Estrada, José Arturo Castillo-Cardiel

**Affiliations:** 1Laboratorio de Investigación en Nutrición, Facultad de Medicina, Universidad Autónoma del Estado de México, Toluca 50180, Mexico; dsmn.amapola@gmail.com (A.D.S.-M.); rvaldesr@uaemex.mx (R.V.-R.); flordemariacruzest@gmail.com (F.d.M.C.-E.); 2Departamento de Ciencias Ambientales, Universidad Autónoma Metropolitana, Lerma 52005, Mexico; j.aguirre@correo.ler.uam.mx; 3Department of Research, Continuing Education and Distance Learning, Universidad Autónoma de Durango, Durango 34209, Mexico; arturocardiel@yahoo.xom.mx

**Keywords:** carbohydrate, body mass index, type 2 diabetes *mellitus*, microbiota

## Abstract

Type 2 diabetes *mellitus* (T2DM) is a multifactorial disease, influenced by dietary and environmental factors that can modify the intestinal microbiota. The aim of this study was to evaluate changes in the composition and diversity of the intestinal microbiota associated with carbohydrate (CHO) consumption in T2DM patients. Forty patients participated, with and without T2DM. Fecal samples were collected for the characterization of microbial diversity from the massive sequencing of the 16S rRNA gene. Carbohydrate consumption was quantified using the Frequency Consumption Foods questionnaire (FCF), the groups were categorized according to Body Mass Index (BMI) and BMI + CHO consumption. The group without T2DM showed normal biochemical and anthropometric parameters, although they had a high carbohydrate consumption compared to the group with T2DM. At the *phylum* level, there were differences in relative abundance; the control overweight group (CL–OW > CHO) and T2DM-Normal Weight > CHO patients had increased *Bacteroides* and decreased *Firmicutes*. In contrast, the CL–OW > CHO and T2DM-OW < CHO patients, showed reduced *Bacteroidetes* and an elevated amount of *Firmicutes*. At the genus level, the differences were in the relative abundance of *Roseburia*, *Clostridium_IV*, *Prevotella*, and *Sporobacter*, associated with the consumption of carbohydrates. The groups that consumed high amounts of carbohydrates, regardless of whether they had diabetes *mellitus* or were overweight, had a significantly reduced proportion of *Faecalibacterium*, an altered proportion of *Bacteroides*. The high consumption of carbohydrates showed considerable modifications in the composition and diversity of the bacterial communities.

## 1. Introduction

Type 2 diabetes *mellitus* (T2DM) occurs when the pancreas does not produce enough insulin or the body does not use it effectively, which causes glucose intolerance [1]. In 2015, there was an estimated 415 million people with DM; by 2040, the number will be 642 million [2]. In Mexico, T2DM is the main cause of morbidity and mortality among individuals aged between 45 and 64 years [3]. It is a multifactorial disease involving genetic, dietary, lifestyle (e.g., sedentarism) [4], and environmental factors (e.g., intestinal microbiota) [5]. Intestinal microbial composition is considered to be a factor of environmental origin because it differs among patients according to their environments [6]. Recent studies propose a link between the composition of the intestinal microbiota, its metabolites and the health of individuals [7,8]. Accordingly, the type, quantity, and quality of dietary elements consumed modify patients’ microbiota, which causes dysbiosis that influences the pathophysiology of T2DM [9]; however, it is still unclear how these mechanisms are articulated. The intestinal microbiota is defined as a complex and diverse community dominated by bacteria living in the intestine [10]. In healthy adults, the most abundant bacterial phyla are *Bacteroidetes* (Gram-negative bacteria) and *Firmicutes* (Gram-positive bacteria) [11]; *Actinobacteria*, *Fusobacterium*, and *Verrucomicrobia* are variable among individuals [12], but their composition can vary in some pathologies. Several studies show that the type of microbes present in the intestine drive and influence some of the metabolic modifications in individuals that cause the presence of obesity and T2DM [13,14,15,16,17,18]. A high-sugar diet is associated with the symptoms of type 2 diabetes in rats, along with inflammation, alterations in glucose metabolism, and a greater abundance of harmful bacteria in the intestine [19]. In addition to this, it has been reported that modifying the composition of the intestinal microbiota improves insulin sensitivity [20] and that a low abundance of *Lactobacillus*, *Prevotella*, *Bacteroides*, *Desulfovibrio*, and *Oxalobacter spp* promotes metabolic disorders such as Insulin Resistance (IR) [21]. However, in patients with T2DM who are not overweight, a reduced abundance of *Akkermansia muciniphila* has been shown, also associated with a reduction in insulin secretion [22,23,24]. Other microorganisms related to metabolic effects in T2DM are *Ruminococcus, Clostridium, Bifidobacterium, Bacteroides, Eubacterium,* and *Listeria* among others [25,26]. The presence of *Blastocystis spp* and *Prevotella copri* are indicators of a favorable postprandial glucose metabolism [27]; therefore, the alteration of their abundance generates a greater risk of developing T2DM with alteration in postprandial glucose metabolism [28].

Each nutrient consumed in the diet alters the composition of the intestinal microbiota, promoting the expansion of microorganisms [29]. Studies have described the effects of high-fat diets [30], but there have been no studies on the impact of consuming different types of carbohydrates (CHOs) on the intestinal microbiota. For example, most individuals consume different proportions of sucrose as a dietary additive [31]. Hydrolyzed sucrose is retained in the distal intestine, exposing the microbiota to significant amounts of fructose and glucose for a longer period [32]. Sucrose has been found to decrease the abundance of *Bacteroides thetaiotaomicron*, which are important regulators of intestinal colonization [33]. Evaluating the nutritional effects of CHO consumption is complex because food content is reported in different ways; for example, starch intake is often reported as total starch intake without specifying the subtype and digestibility [34]. Another factor is that different terms can be used to describe the same CHO; for example, sugars can refer to sugars, total sugars, total available sugars, free sugars, added sugars, refined sugars, simple sugars, or caloric sweeteners. This makes it impossible to assign a specific effect to a specific CHO, leading to a confusing nutritional interpretation [35]. Therefore, it is difficult to evaluate and associate the consumption of a specific type of CHO with its nutritional impact on the microbiota and metabolism of individuals [5,36]. Although several studies have evaluated the effects of T2DM on the body [37,38,39], studies evaluating the effects of CHO consumption associated with T2DM on the body are scarce. Some animal models have shown significant changes in the peripheral immune system [5,40] and mucous membranes in the small intestine [41,42], but there has been little evaluation of this relationship in humans.

Thus, the objective of this study was to evaluate the changes in the composition and diversity of the intestinal microbiota associated with CHO consumption in T2DM patients.

## 2. Results

### 2.1. Sociodemographic Data Obtained from Questionnaires Given to Patients with and Without T2DM

The median age in the T2DM group was higher (49 years) than that in the control (CL) group (42 years). In the T2DM subgroups, the T2DM–overweight (OW) subgroup was younger (median age, 47 years) than the T2DM–normal weight (NW) subgroup (median age, 50 years). In the CL-OW and CL-NW subgroups, the median ages were 36 and 45 years, respectively. At the time of the study, all patients in the CL group were healthy, and the patients with T2DM did not have aggregate pathologies recorded. Each group comprised eighteen women and two men. In the T2DM group, a total of twelve patients had a 6- to 10-year evolution of T2DM, whereas only eight had less than 5 years of diagnosis. The treatment most used by patients was metformin (40%), followed by insulin (25%), a metformin plus glibenclamide combination (20%), an insulin plus metformin combination (10%), or a sitagliptin plus metformin combination (5%).

### 2.2. Anthropometric Evaluation of Patients with and Without T2DM

The body mass index (BMI) was 24.75 kg/m^2^ for the CL group and 24.8 kg/m^2^ for the T2DM group. The BMIs of the patients were categorized, with the results showing twelve patients with NW and eight OW patients in both the CL and T2DM groups.

### 2.3. Biochemical Evaluation of Glycemia, Hemoglobin A1C, and Triglycerides of Patients with and Without T2DM

The biochemical parameters of glycemia and hemoglobin A1c (HbA1c) were found to be significantly elevated (*p* < 0.001) in the T2DM group compared to the CL group (Table 1). Patients in the CL-OW subgroup showed elevation in both glucose and HbA1c compared with those in the CL-NW subgroup. By contrast, the T2DM-NW subgroup of patients showed elevated glucose and HbA1c compared to the T2DM-OW subgroup (Table 1). Regarding triglycerides, a difference was found between the CL group and the T2DM group (*p* < 0.024); the T2DM-OW subgroup showed higher levels of triglycerides than the T2DM-NW, CL-OW, and CL-NW subgroups (Table 1). Total cholesterol was lower in the CL group compared to the T2DM group (*p* < 0.029); however, there were no significant differences between subgroups (*p* < 0.519; Table 1).

### 2.4. Dietary Assessment with the Food Frequency Questionnaire

The dietary evaluations of the groups showed significant differences (*p* < 0.009), which were assessed with the Food Frequency Questionnaire (FFQ). The T2DM group consumed less energy (1266 kcal/day) with an adequate proportion of CHO (113 g/day), fiber (13 g/day), protein (60 g/day), and lipids (34 g/day) compared to patients without DM, who consumed more energy (2284 kcal/day), CHO (249 g/day), dietary fiber (25 g/day), proteins (86 g/day), and lipids (66 g/day). The CL group consumed more CHOs such us sucrose (22 g/day), starch (18 g/day), and fructose (8 g/day), with the exception of glucose (3.5 g/day), compared to the T2DM group who consumed 30 g/day of starch, 25 g/day of sucrose, the same proportion of fructose (8.6 g/day), and a higher amount of glucose (6 g/day).

### 2.5. Microbiota Analysis of Patients with and Without T2DM

The richness and diversity of operational taxonomic units (OTUs) were estimated using the Chao1 index as an indicator of richness, the Shannon and inverse Simpson’s indices as indicators of diversity, and the Pielou index as an indicator of equity (Table 2). Using these approaches, we found that the fecal bacterial communities in the T2DM-NW subgroup showed lower values than those obtained for the T2DM-OW subgroup. This contrasted with the observed in the CL-OW subgroup, which had a greater richness and diversity than the T2DM-OW subgroup. The percentage of coverage was very similar in all samples from the different subgroups (Table 2). The results of massive sequencing targeting the V4–V5 region of the 16S rRNA gene of the fecal samples were used for taxonomic characterization at different levels. An analysis of the relative abundance at the phylum and sex levels was performed in the subgroups categorized by BMI and BMI + CHO. The analysis of the relative abundance at the phylum level showed that the taxonomic distribution of the BMI and BMI + CHO subgroups had a higher abundance of *Firmicutes* and *Bacteroidetes*, followed by *Proteobacteria, Actinobacteria*, and *Verrucomicrobia*. In groups categorized according to BMI, *Firmicutes* were most highly represented in the CL-OW subgroup (70.62%) and *Bacteroidetes* in the T2DM-OW subgroup (54%). *Proteobacteria* had higher representation in the CL-NW subgroup (0.95%), whereas *Actinobacteria* did not reach a representation greater than 0.5% in any of the subgroups. However, the subgroups with the highest representation were T2DM-NW (0.27%) and CL-NW (0.26%); *Verrucomicrobia* (1.20%) was mostly represented in the T2DM-NW subgroup (Figure 1). In the BMI + CHO subgroups, *Firmicutes* were mostly represented in the CL-OW < CHO subgroup (75.30%). This was in contrast to *Bacteroidetes*, which had a higher representation in the T2DM-OW < CHO subgroup (54%), followed by T2DM-NW > CHO (49%). *Proteobacteria* showed greater representation in the T2DM-NW > CHO (1.16%) and CL-NW < CHO (1.04%) subgroups. *Actinobacteria* were mostly represented in the CL-NW > CHO subgroup (0.65%) and their levels were very low in the other subgroups, not reaching 0.5%, whereas *Verrucomibrobia* (1.43%) was most highly represented in the T2DM-NW < CHO subgroup.

The evaluation of relative abundance at the genus level according to BMI and BMI + CHO showed that the taxonomic distribution had a greater abundance of 25 taxa, with the results of the CL-NW subgroup used as a reference. Categorization of the subgroups based on BMI (Figure 1) revealed the following findings: CL-NW subgroup: *Faecalibacterium* and *Bacteroides*; CL-OW subgroup: *Megamones* and *Catenibacterium*; T2DM-NW subgroup: *Lachnospiracea_incertae_sedis* and *Ruminococcus*; and T2DM-OW subgroup: *Prevotella* and *Clostridium_IV*. Categorization of the subgroups according to BMI + CHO revealed the following findings: CL-NW > CHO*: Faecalibacterium* and *Roseburia*; CL-NW < CHO: *Streptococcus* and *Dorea*; CL-OW > CHO: *Blautia* and *Ruminococcus*; CL-OW < CHO: *Megamones* and *Catenibacterium*; T2DM-NW > CHO: *Bacteroides* and *Coprococcus*; T2DM-NW < CHO: *Lachnospiracea_incertae_sedis* and *Lactobacillus*; and T2DM-OW < CHO: *Prevotella* and *Clostridium_IV* (Table 3).

Different taxonomic groups were identified at the sex level with statistical significance according to the condition of BMI + CHO. The results showed that four taxa had statistical significance: *Roseburia* (*p* = 0.025), *Clostridium_IV* (*p* = 0.017), *Prevotella* (*p* = 0.017), and *Sporobacter* (*p* = 0.020). To test for differences in bacterial composition between the samples, principal component analysis (PCA) was performed for all genus-level 16S rRNA gene reads suing the Statistical Analysis of Metagenomic Profiles with STAMP software (STAMP v2.1.3 setup package for Microsoft Windows, is open source, https://beikolab.cs.dal.ca/software/STAMP, accessed on 12 November 2024). In this analysis, the differences that existed between the dominance of the bacterial communities and the characteristics of each subgroup based on BMI and BMI + CHO were observed (Figure 2a). In the BMI group, the T2DM-OW subgroup showed a clearly differentiated position from that of the other subgroups. By contrast, in the BMI + CHO group, the T2DM-OW < CHO subgroup was separated from the other subgroups (Figure 2b).

The data are presented as the means ± standard deviations of the biochemical values of patients with and without DM. A one-way analysis of variance (ANOVA) was performed to analyze differences between groups, with Tukey’s post hoc test used to compare intragroup differences. *p* > 0.05 was considered statistically significant.

## 3. Discussion

The dietary evaluation carried out in this study showed a higher CHO consumption in the healthy CL group than in patients with T2DM. This phenomenon acts as a risk factor for the healthy group but is protective for the diabetic group. For example, the dietary recommendation for patients with T2DM establishes a CHO consumption of less than 45%, and a sugar consumption of less than 10%/day of the total energy consumed [43]; a sucrose consumption below 5% has also been proposed [44]. In this study, both groups consumed more than 50% CHO and 10% sucrose in their diet, a situation that can be associated with being OW [45]. The richness and diversity of the bacterial communities in the feces samples of the subgroups categorized by BMI and BMI + CHO (Table 2 and Figure 2) allowed us to observe the modifications in the structure of the bacterial communities of these subjects. In the T2DM-NW > CHO subgroup, the richness was much lower compared to that of the CL-NW > CHO subgroup, while in the CL-OW < CHO subgroup, there was greater richness than in the T2DM-OW < CHO subgroup (Table 2). These results showed the changes in diversity of the intestinal microbiota in patients who were OW, had T2DM, and had a high CHO consumption. Similar behavior has been reported in OW patients, who tend to be obese and present with changes in the diversity of the intestinal microbiota, which leads to low-grade inflammation [46], and consequently, in the long term, may be the cause of T2DM [47]. It is known that the type of intestinal microbiota depends on the quality and quantity of the nutrients consumed, particularly CHO. Patients with T2DM who take metformin for glycemic control reportedly experience changes in their intestinal microbiota [48]. The results from this study suggest that the consumption of CHO and the status of being OW alter the structure of the intestinal microbiota of patients with T2DM [49].

Characterization of the bacterial communities in the feces showed that the relative abundance of Firmicutes in the T2DM-OW subgroup was decreased by almost 50% of the total (Firmicutes/Bacteroidetes ratio) compared to the CL-OW group, where the percentage of Firmicutes reached 70.62% (Figure 1). These findings are in contrast to the results of the study by Chávez-Carbajal et al. [50], which showed that the T2DM + metformin group had an increase in *Bacteroidetes* and a decrease in *Firmicutes*, allowing for the association of metformin with the modification of the microbiota. In addition, an interesting finding was that the percentages of abundance in the T2DM-OW group (*Bacteroidetes* 54% and *Firmicutes* 45.23%) in this study (Figure 2a,b) were similar to those of the T2DM group without metformin (*Bacteroidetes* 53.18%/*Firmicutes* 43.11%) in the study by Chávez-Carbajal et al. [50]. Thus, there may be a close relationship between OW, T2DM, metformin consumption, and changes in the intestinal microbiota. Similarly, other studies have shown a clear relationship between OW/obesity, T2DM [51], and the diversity of *Bacteroidetes* versus *Firmicutes* [9], where patients with T2DM and obesity have a lower percentage of *Bacteroidetes* (19.5%) and a higher percentage of *Firmicutes* (55.7%) compared to patients without T2DM (*Bacteroidetes*: 32.1%/*Firmicutes*: 36.9%) [52]. The importance of maintaining an NW in patients suffering from T2DM is clear, as being OW/obese can affect the therapeutic effects of metformin [52,53]. When the subgroups of BMI + CHO are combined, it can be inferred that T2DM combined with a high CHO consumption could be the main cause of the changes in the abundance of *Firmicutes*. This was observed in the T2DM-NW > CHO subgroup (49.72%), in which there were less Firmicutes compared to the CL-OW > CHO subgroup (56.59%). The level of *Firmicutes* was higher in the CL-OW < CHO subgroup (75.30%), with a combination of OW and low CHO consumption without T2DM, and was lower in the T2DM-NW < CHO subgroup (67.25%), with a combination of NW and low CHO consumption with T2DM. OW patients with T2DM and low CHO consumption (T2DM-OW < CHO, 45.23%) showed a lower percentage of *Firmicutes* (Figure 1), consistent with published data regarding dysbiosis attributed to *Firmicutes* in the intestinal microbiota of women [50]; interestingly, 90% of patients in this study were women. Additionally, it has been reported that *Firmicutes* can contribute to the development of obesity [54], in accordance with the results of this study, as there is an abundance of *Firmicutes* associated with OW/obesity in Mexican women. On the other hand, there was greater prevalence of *Bacteroidetes* in the T2DM-OW < CHO subgroup (54%), and a lower proportion of *Proteobacteria*, *Actinobacteria*, and *Verrucomicrobia* (Figure 1). It is believed that *Bacteroidetes* and *Actinobacteria* may inhibit the growth of *Firmicutes* [55]. The present study found that there was a higher relative abundance of *Bacteroides* (29.75%) than other genera in the T2DM-NW > CHO subgroup. Changes in the abundance of *Bacteroides* could be attributed to the amount consumed versus absorbed; for example, sucrose, which is not absorbed in the small intestine, is absorbed in the colon, where it also interacts with the colonic microbiota [56]. This has also been reported in Japanese patients with T2DM [57]. These results are controversial because the ingestion of sucrose produces a reduction in *Bacteroides* by inhibiting the expression of the BT3172 gene, which is essential for its survival [33,58,59].

Another genus associated with a high prevalence of prediabetes [60] and childhood obesity [61] is *Megamones*, which showed a high prevalence in the CL-OW < CH subgroup in this study. In addition, *Megamones* was found to be highly enriched in a group with normal glucose tolerance [62], revealing that the different species of *Megamones* have particular functions; some are capable of fermenting glucose into acetate and propionate, and short-chain fatty acids, which are beneficial to health [63].

There was a higher relative abundance of the genus *Prevotella* in the T2DM-OW < CHO subgroup. *Prevotella* is a bacterium associated with chronic intestinal inflammation, which is not abundant in Chinese patients with T2DM [64]. In this study, the abundance analysis performed between subgroups identified *Prevotella* in the group of most abundant intestinal bacteria. The abundance of these bacteria increases in subgroups with T2DM, making dysbiosis evident in these patients [6,9,47]. In addition, inflammation of the intestinal mucosa mediated by this bacterium promotes systemic inflammation, with increased intestinal permeability and translocation of bacterial products [65,66]. The genus *Faecalibacterium* had a higher relative prevalence in the CL-NW > CHO subgroup and a lower relative prevalence in the T2DM-NW > CHO subgroup. *Faecalibacterium* is one of the most abundant bacterial species in the healthy human intestine [67]. Currently, *F. prausnitzii* is the only known species of *Faecalibacterium* that, in recent years, has been of great interest as a biomarker of intestinal health [68]. The decrease in the abundance of this bacterial genus is associated with inflammatory bowel syndrome, inflammatory bowel disease, colorectal cancer, and T2DM [69]. However, the effects derived from the interaction between *F. prausnitzii* and the consumption of CHO in patients with T2DM are not known. There are variations in the microbial composition of the subgroups categorized by BMI and BMI + CHO. At the genus level, a clear bacterial profile was observed within the T2DM-OW and T2DM-OW < CHO subgroups. This finding would help confirm the presence of a microbial signature in T2DM, as previously reported [56]. Therefore, this study could serve as a basis for identifying the microbial molecular fingerprint of the colon in patients with T2DM, OW, and CHO consumption. The fact that the intestinal microbiota is a dynamic ecosystem that changes with diet [29] could explain the modifications found with the consumption of CHO in this study. CHO consumption significantly marked the abundance of certain bacteria in the different subgroups including BMI + CHO. At the sex level, there was a significant increase in *Roseburia* and *Clostridium IV* (CL-NW > CHO), as well as *Prevotella* (T2DM-OW < CHO) and *Sporobacter* (T2DM-NW > CHO). These findings provide insights into the effects and interaction between sucrose consumption and the changes it causes in the host’s intestinal microbiota [12,32,70]. In the colon, the interaction between the microbiota and the products of sucrose metabolism [55] can influence the inflammatory state and function of the intestinal barrier, and thus, the development of complications in T2DM. This study may lead to the proposal of new nutritional education strategies, which favor a reduction in CHO in the diet, and accordingly, modulate the various bacterial groups associated with the development of TDM2.

## 4. Materials and Methods

### 4.1. Ethics Statement

The present study protocol was reviewed and approved by the ethics committee of the hospital and the Faculty of Medicine (project 015/2018) of the Universidad Autónoma del Estado de México (UAEMéx). All participants included in the study gave their informed consent in accordance with the Declaration of Helsinki, revised in 2013 [71].

### 4.2. Study Design

This was a prospective, cross-sectional, and comparative study. One hundred subjects with and without a diagnosis of T2DM were invited to attend an informational talk at the hospital’s outpatient clinic. From this group, 40 subjects of both sexes attended the outpatient clinic of the General Hospital “Dr. Nicolás San Juan” in Toluca, State of Mexico. A brief medical history was given to assess whether the patients met the inclusion criteria.

### 4.3. Study Subjects

Of the 100 subjects invited to participate, 40 met the inclusion criteria and were divided into two groups: the T2DM group (*n* = 20) and a CL group of healthy subjects without DM (*n* = 20). The inclusion criteria for both groups were 25 to 65 years old, without any other pathology at the time of the study, and less than 10 years of struggling with the disease, regardless of chronological age. The following exclusion criteria were used in both cases: pregnancy; chronic alcohol, drug, or tobacco use; and acute or chronic autoimmune, bacterial, or viral diseases.

### 4.4. Anthropometric Evaluation

To calculate the BMI, each participant’s body weight in kilograms was determined using the Tanita BF-682 Body Fat Monitor Scale (Monterrey, Mexico) after an 8 h fast. The participants were asked to stand on the scale, with the soles of their feet on the scale’s surface. Height was measured while standing, without shoes, with a stadiometer (Seca^®^ Model 240; accuracy ± 2 mm; Seca GmbH & Co., Hamburg, Germany). Once these data were obtained, BMI was calculated using the following formula: weight (kg)/height (m^2^). Once the BMI was obtained, it was categorized as follows: NW (18.5–24.99 kg/m^2^), OW (25.0–29.99 kg/m^2^), and obese > 30 kg/m^2^.

### 4.5. Determination of Biochemical Profile

To obtain the blood samples, the patients were asked to fast for 8 h prior to collection of the sample. Whole venous blood was collected in tubes with heparin (Vacutainer; Becton, Dickinson and Company, Franklin Lakes, NJ, USA). Glycemia, glycosylated hemoglobin, total cholesterol, and triglycerides were determined from whole blood using immunoturbidimetric tests with reinforcing particles (Innovastar; Diasys Diagnostic Systems, Holzheim, Germany).

### 4.6. Dietary Evaluation and Patient Categorization

The FFQ was used to evaluate the food consumed in the last month. Based on this information, the energy intake of macronutrients (CHO, lipids, and proteins) was calculated for each patient, with the help of two instruments: The Mexican List of Composition of Food Equivalents “Mexican System of Equivalents” [72] and the FoodData Central, U.S. Department of Agriculture (https://fdc.nal.usda.gov/ accessed on 12 November 2024) [73]. From the intake in grams per day of CHOs, a consumption classification was made of more than 200 g/day (>200) or less than 200 g/day (<200), considering the proportions of sucrose, fructose, glucose, and maltose. With these data, the groups were categorized by BMI and subcategorized according to BMI + CHO consumption, as shown in Figure 3. 

### 4.7. Collection of Feces

To collect feces samples, a sterile bottle was provided with instructions for correct collection and storage. Once the samples were collected, they were processed in a sterile microbiological hood. Four aliquots of 1 g of feces were stored in sterile 1.5 mL microtubes at −70 °C for subsequent batch analysis [50].

### 4.8. Analysis of Fecal Microbiota

(1)DNA extraction from feces

The 40 samples from the study participants were processed, using 150 mg feces from each patient for DNA extraction, following the instructions provided in the protocol of the Quick-DNA™ Fecal/Soil Microbe Miniprep Kit (Zymo Research, Irvine, CA, USA). The length of the mechanical lysis with a disruptor was modified from 10 to 15 min.

(2)Gel electrophoresis of DNA extraction and metagenomics’ material amplification test (PCR)

Electrophoresis was performed at 110 V for 35 min on 3 µL DNA loaded on 1% agarose gels (100 mL of 1× Buffer TAE per 1 g agarose). A total of 1 mL DNA from each sample was used to amplify by PCR targeting the 16S rRNA gene, to verify that it was capable of being amplified [74].

(3)Massive sequencing

(3.1) Illumina sequencing by amplicons of the V4–V5 region of the 16s rRNA gene

The 40 samples resulted in metagenomics DNA that met the quality of purity and concentration (260/280 = A1.8–2.0). Massive Illumina MiSeq sequencing (300 + 300 bp PE) [75] was conducted at the Center for Comparative Genomics and Evolutionary Bioinformatics of Dalhousie University (Halifax, Canada).

(3.2) Bioinformatics analysis of the sequences

The 16S rRNA data were processed with Mothur software version 1.39.5, following the pipeline recommended by the developer of the MiSeq SOP. The selected reads met the following criteria: no ambiguous bases, a minimum length of 200 bp, and homopolymers with a maximum length of 8 bp. Similarly, removal of the chimeric sequences and the lineages of eukaryotes, mitochondria, and chloroplasts, unknown and unclassified, was carried out. Subsequently, the diversity of the sequences of OTUs was examined considering quality readings with a dissimilarity of 3%, while the rarefaction curves were calculated with a similarity of 97% with the Mothur alpha diversity flow. Different metrics were calculated to evaluate the bacterial communities, including the number of OTUs observed, the Shannon diversity indices and Pielou uniformity, and the Chao1 and inverse Simpson estimators for species richness. The calculation of these parameters was carried out by normalizing all of the libraries. Finally, the composition of the bacterial communities was determined according to beta diversity. The abundance was expressed as a percentage with respect to the total number of sequences in each sample (relative abundance) [76,77]. Subsequently, the file generated by MOTHUR was analyzed using Statistical analysis of taxonomic and functional profiles (STAMP software, STAMP v2.1.3 setup package for Microsoft Windows, is open source, https://beikolab.cs.dal.ca/software/STAMP, accessed on 12 November 2024) and the results are represented by graphs. The obtained sequences were registered in the NCBI BioSample database, at the following link: https://www.ncbi.nlm.nih.gov/sra/PRJNA807457, accessed on 12 November 2024), with the number to access and cite these SRA data of PRJNA807457.

### 4.9. Statistical Analyses

The anthropometric, biochemical, and diet profiles were analyzed using SPSS software version 22 (IBM, Armonk, NY, USA), employing the non-parametric Kruskal–Wallis test. To identify the differences between the subgroups of the intestinal microbiota at the taxonomic levels of phylum, class, order, and genus, the relative abundance (%) of bacterial communities, we used ANOVA and the Kruskal–Wallis test to calculate *p* < 0.05 in STAMP [75,77]. 

## 5. Conclusions

The composition and diversity of the gut microbiota in patients with and without T2DM were significantly different. The habitual consumption of CHOs is closely related to differences in taxonomy at the phylum and genus levels. Although more studies are necessary, it can be concluded that, with a reduction in CHO consumption to 10% per day, it is possible to modulate the bacterial communities in healthy patients, which may help prevent dysbiosis and the development of T2DM. However, with an increase in CHO consumption (>15%), the modification in proportion and diversity of the microbiota was altered. The groups that consumed high amounts of CHOs, regardless of whether they had T2DM or were OW, had a significantly reduced proportion of *Faecalobacterium*, an altered proportion of *Bacteroides*, and important modifications to other bacterial genera. The high consumption of CHOs considerably modified the composition and diversity of the bacterial communities. If comorbidities such as T2DM and OW are added to this analysis, the changes in the intestinal microbiota become more evident. In this study, CHO consumption showed, at the genus level, an increase in the abundances of bacteria such as *Faecalobacterium, Bacteroides, Blautia, Roseburia*, *Prevotella*, and *Sporobacter*. 

## Figures and Tables

**Figure 1 ijms-25-12359-f001:**
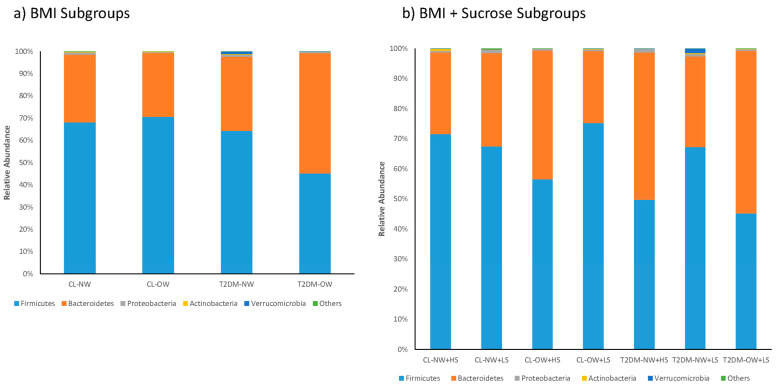
Study groups: control and type 2 diabetes mellitus, classified by body mass index (BMI) and subcategorized by BMI + CHO consumption. Structure bacterial communities of the feces samples of the CL and T2DM groups and their respective subgroups according to BMI and BMI + Sucrose. (**a**) Average relative abundance at the *Phylum* level of the BMI subgroups. (**b**) Average relative abundance at the phylum level of the BMI + Sucrose subgroups. Only taxa with mean relative abundances > 0.05% were graphed. Control - Normal Weight (CL-NW), Control-Normal Weight + High Sucrose (CL-NW + HS), Control-Normal Weight +Low Sucrose (CL-NW + LS), Control-Overweight (CL-OW), Control-Overweight + High Sucrose (CL-OW + HS), Control-Overweight +Low Sucrose (CL-OW + LS), Type 2 Diabetes *Mellitus*–Normal Weight (T2DM-NW), Type 2 Diabetes *Mellitus*-Overweight (T2DM-OW), Type 2 Diabetes *Mellitus*–Normal Weight +High Sucrose (T2DM-NW + HS), Type 2 Diabetes *Mellitus*–Normal Weight + Low sucrose (T2DM-NW + LS), Type 2 Diabetes *Mellitus*–Overweight (T2DM-OW), Type 2 Diabetes *Mellitus*-Overweight + Low Sucrose (T2DM-OW + LS).

**Figure 2 ijms-25-12359-f002:**
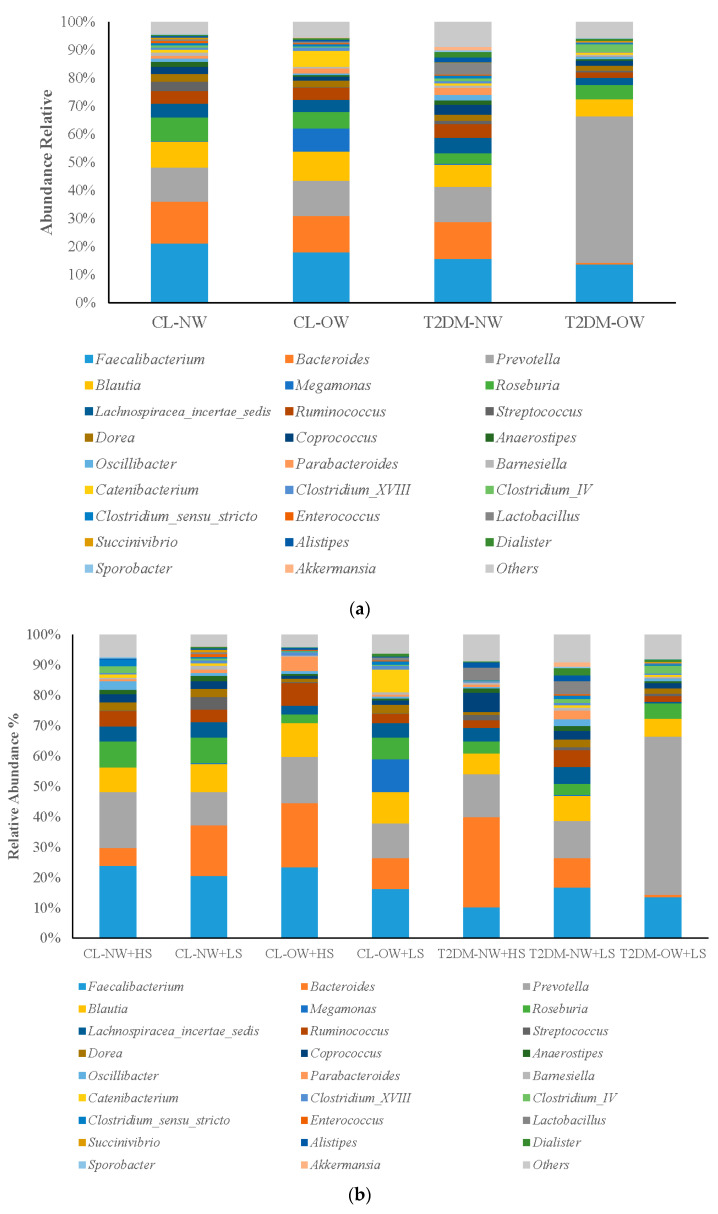
(**a**) Structure of the bacterial communities in the feces samples from the CL and T2DM groups and their respective subgroups according to BMI. Average relative abundance at the *phylum* level in the BMI subgroups. Only taxa with mean relative abundances > 0.05% were graphed. Control normal weight (CL-NW), control overweight (CL-OW), type 2 diabetes *mellitus* normal weight (T2DM-NW), Type 2 diabetes mellitus overweight (T2DM-OW). (**b**) Structure of the bacterial communities in the feces samples from the CL and T2DM groups, with the subgroups categorized by BMI. Average relative abundance at the genus level in the BMI subgroups. Taxa with mean relative abundance *p* > 0.05% were plotted. Control normal weight (CL-NW), control overweight (CL-OW), type 2 diabetes mellitus normal weight (T2DM-NW), type 2 diabetes mellitus overweight (T2DM-OW).

**Figure 3 ijms-25-12359-f003:**
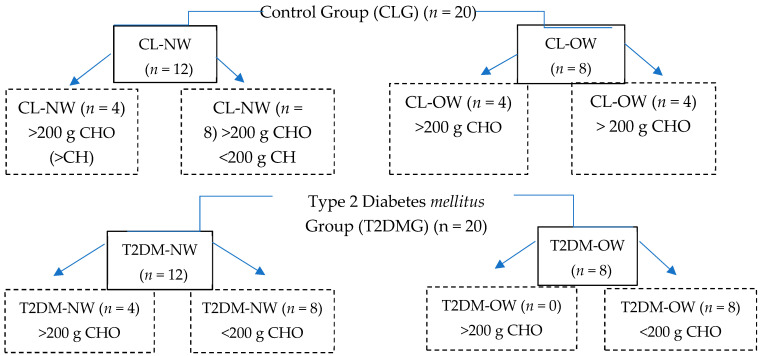
Classification of groups by body mass index and subgroups categorized by body mass index plus carbohydrate consumption. Carbohydrate (CHO), control normal weight (CL-NW), control overweight (CL-OW), type 2 diabetes *mellitus* normal weight (T2DM-NW), type 2 diabetes *mellitus* overweight (T2DM-OW).

**Table 1 ijms-25-12359-t001:** Biochemical data of patients with and without diabetes *mellitus*.

	CL-NW *n* = 12	CL-OW *n* = 8	T2DM-NW *n* = 12	T2DM-OW *n* = 8	
	Mean ± SD	Mean ± SD	Mean ± SD	Mean ± SD	*p* Value
Serum glycaemia mg/dL	81.1 ± 1.7	83.3 ± 5.9	150 ± 7.1 **	135 ± 4.3 **	0.001 *
Hb1Ac %	5.1 ± 0.206	5.4 ± 0.471	9.2 ± 1.5 **	8.4 ± 3.3 **	0.001 *
TG mg/dL	128 ± 3.6	155 ± 8.3	208 ± 1.5	251 ± 1.5	0.119
CT mg/dL	175.2 ± 2.3	174 ± 2.4	195 ± 6	192 ± 3.9	0.519

The data represent the means ± standard deviations of the biochemical values of patients with and without diabetes *mellitus*. One-way ANOVA * was performed to analyze differences between groups, with Tukey’s post hoc ** test to compare intragroup differences. Significance was considered with a *p* > 0.05. Control normal weight (CL-NW), control overweight (CL-OW), type 2 diabetes *mellitus* normal weight (T2DM-NW), type 2 diabetes *mellitus* overweight (T2DM-OW).

**Table 2 ijms-25-12359-t002:** (**a**) Number of qualified readings, estimators of richness, species, diversity, and uniformity of the bacterial communities present in samples of patients without T2DM categorized by BMI. (**b**) Number of qualified readings, estimators of richness, species, diversity, and uniformity of the bacterial communities present in stool samples of patients with T2DM categorized by BMI.

a.							
Control Group	Nseqs ^a^	Coverage ^b^ (%)	OUT ^c^	Chao1 (lci ^d^-hci ^e^)	Inv-Simpson (lci ^d^-hci ^e^)	Shannon (lci ^d^-hci ^e^)	Pielou
	**CL-NW (*n* = 12)**						
**M15**	25,575	99.75	278	412 (357–520)	6 (5.94–6.12)	2.81 (2.8–2.83)	0.993
**M45**	55,419	99.82	437	594 (572–638)	6.27 (6.21–6.34)	4.75 (4.72–4.79)	0.991
**M13**	58,542	99.86	612	857 (795–97)	4.74 (4.70–4.78)	3.45 (3.44–3.47)	0.994
**M23**	57,984	99.86	1029	1297 (1245–1391)	6.44 (6.41–6.47)	4.44 (4.43–4.45)	0.998
**M29**	98,637	99.86	906	1264 (1162–1374)	5.29 (5.26–5.31)	4.12 (4.11–4.13)	0.997
**M32**	56,648	99.86	1125	1468 (1436–1666)	4.39 (4.35–4.44)	3.10 (3.09–3.11)	0.998
**M33**	45,032	99.88	832	1160 (1059–1363)	5.73 (5.71–5.75)	4.60 (4.59–4.61)	0.997
**M34**	64,036	99.83	1081	1396 (1344–1508)	6.55 (6.5–6.6)	4.99 (4.98–5)	0.996
**M37**	50,042	99.86	458	597 (505–641)	4.66 (4.62–4.71)	2.46 (2.45–2.47)	0.996
**M38**	5021	99.66	1191	1490 (1488–1688)	6.52 (6.38–6.67)	4.86 (4.82–4.87)	0.998
**M43**	46,370	99.87	883	1239 (1217–1288)	6.03 (5.98–6.09)	4.60 (4.58–4.61)	0.997
**M46**	39,738	99.83	950	1290 (1188–1334)	6.32 (6.23–6.40)	4.55 (4.53–4.56)	0.998
	**CL-OW (*n* = 8)**						
**M48**	6333	99.82	1146	1503 (1471–1704)	5.59 (5.51–5.68)	4.75 (4.73–4.76)	0.997
**M49**	14,430	99.83	998	1330 (1318–1434)	5.41 (5.36–5.45)	4.85 (4.84–4.88)	0.994
**M6**	8343	99.75	1037	1345 (1253–1410)	5.71 (5.63–5.8)	4.84 (4.82–4.86)	0.997
**M16**	59,311	99.85	681	908 (849–1009)	5.99 (5.95–6)	4.9 (4.8–5)	0.979
**M17**	52,056	99.84	1015	1332 (1267–1450)	4.87 (4.82–4.92)	3.43 (3.42–3.44)	0.997
**M19**	40,529	99.81	861	1128 (1072–1229)	6.14 (6–6.2)	4.75 (4.74–4.76)	0.999
**M25**	64,872	99.88	822	1091 (1080–1202)	5.43 (5.41–5.45)	4.59 (4.58–4.61)	0.778
**M30**	21,765	99.74	625	884 (872–919)	4.94 (4.86–5)	2.14 (2.13–2.16)	0.992
**b.**							
**T2DM** **Group**	**Nseqs** **^a^**	**Coverage** **^b^** **(%)**	**OUT** **^c^**	**Chao1** **(lci ^d^-hci ^e^)**	**Inv-Simpson** **(lci ^d^-hci ^e^)**	**Shannon** **(lci ^d^-hci ^e^)**	**Pielou**
			**T2DM-NW (*n* = 12)**				
**DM5**	44,662	99.86	715	944 (897–1037)	5.65 (5.6–5.7)	4.3 (4.29–4.32)	0.995
**DM38**	28,103	99.84	1130	1471 (1431–1559)	5.62 (5.54–5.7)	4 (4.05–4.08)	0.995
**DM1**	42,462	99.81	703	890 (830–998)	6.24 (6.17–6.32)	3.47 (3.46–3.48)	0.997
**DM2**	63,930	99.84	1047	1346 (1276–1454)	5.98 (5.92–6)	3.53 (3.52–3.54)	0.997
**DM6**	51,875	99.86	879	1172 (1125–1257)	3.77 (3.74–3.8)	3 (3–3.03)	0.995
**DM8**	53,503	99.85	790	1057 (993–1175)	4.27 (4.22–4.31)	3.89 (3.88–3.9)	0.997
**DM9**	63,090	99.82	908	1168 (1090–1299)	4.62 (4.56–4.68)	4.53 (4.52–4.54)	0.999
**DM17**	37,353	99.83	1187	1517 (1460–1629)	6.53 (6.46–6.61)	4.87 (4.86–4.88)	0.998
**DM19**	10,255	99.74	738	1097 (1073–1161)	4.72 (4.6–4.86)	3.95 (3.92–3.97)	0.994
**DM27**	33,597	99.79	680	943 (866–1093)	7.1 (7–7.19)	5 (5.01–5.03)	0.998
**DM50**	40,129	99.82	931	1278 (1203–1425)	6.95 (6.88–7)	4 (4.06–4.08)	0.998
**DM51**	38,123	99.81	799	1056 (981–1198)	6.64 (6.59–6.7)	4.48 (4.47–5)	0.894
			**T2DM-OW (*n* = 8)**				
**DM16**	46,862	99.84	1009	1328 (1275–1425)	5.37 (5.32–5.43)	4.82 (4.81–4.83)	0.999
**DM20**	67,321	99.86	799	1125 (1052–1253)	4.87 (4.84–4.9)	3.33 (3.32–3.34)	0.996
**DM21**	58,523	99.85	1250	1609 (1531–1749)	6.23 (6.18–6.28)	5.23 (5.22–5.24)	0.997
**DM22**	58,044	99.87	1011	1390 (1333–1495)	4.85 (4.81–4.9)	4 (4.06–4.08)	0.997
**DM23**	39,135	99.84	981	1270 (1220–1368)	5.57 (5.52–5.63)	4.75 (4.74–4.77)	0.996
**DM28**	37,674	99.85	922	1366 (1327–1443)	4.75 (4.69–4.82)	2.92 (2.9–2.93)	0.996
**DM42**	18,130	99.74	933	1228 (1166–1367)	5.14 (5–5.25)	3.84 (3.82–3.86)	0.994
**DM42**	71,836	99.86	646	879 (809–1000)	5.96 (5.92–6)	4 (4.04–4.06)	0.998

The parameters were estimated using MOTHUR. Samples from patients without T2DM (M). The parameters were estimated using MOTHUR. Samples from patients with T2DM (DM). **^a^ Nseqs** = sequence numbers that were obtained after removing low quality sequences (N ≥ 2; homopolymers ≥ 8) and short sequences (<200 bp). **^b^** The smallest library (5021 sequences) was used for normalization of the data. The results presented are an average of 1000 repetitions. **^c^ OTU** = number of operational taxonomic units defined over the maximum distance of 3%. **^d^ lci** = lower confidence interval. **^e^ hci** = upper confidence interval.

**Table 3 ijms-25-12359-t003:** Average relative abundance at the genus level, categorized for consumption of carbohydrates *per* group.

	CLNW *n* = 12	%	CLOW *n* = 8	%	T2DMNW *n* = 12	%	T2DMOW *n* = 8	
**>200 g HCO**	*Faecalibacterium* *Prevotella* *Roseburia* *Blautia* *Roseburia* *Bacteroides* Less than 5%	23.92 18.48 8.47 8.11 8.47 5.85 26.7	*Faecalibacterium* *Bacteroides* *Prevotella* *Blautia* *Ruminococcus Parabacteroides* Less than 5%	23.3 21.23 15.23 11.09 7.5 4.83 16.82	*Bacteroides* *Prevotella* *Faecalibacterium* *Roseburia* *Blautia* Less than 5%	29.75 14.19 10.13 8.47 6.82 30.64	No patient consumed more than 200 g of CHO per day, so we do not have samples for this group.	--
	***n* = 4**		***n* = 4**		***n* = 4**		***n* = 0**	**--**
**<200 g HCO**	*Faecalibacterium* *Bacteroides* *Prevotella* *Blautia* *Roseburia* *Streptococcus* Less than 5%	20.49 16.79 10.87 9.32 8.36 4.10 15.67	*Faecalibacterium* *Roseburia* *Prevotella* *Megamones* *Bacteroides* *Blautia* Less than 5%	16.3 15.52 11.41 10.89 10.3 10.28 25.3	*Faecalibacterium* *Prevotella* *Roseburia* *Bacteroides* *Blautia* *Lachnospiracea_* *incertae_sedis* Less than 5%	16.63 12.15 12.1 9.85 8.22 5.71 35.34	*Prevotella* *Faecalobacterium* *Blautia* *Roseburia* Less than 5%	52.13 13.57 5.94 5.03 23.33
	***n* = 8**		***n* = 4**		***n* = 8**		***n* = 8**	

The table shows the percentage and type of bacteria at the genus level classified by the groups of patients who consumed more than 200 g of CH or less than this amount. A percentage of less than 5% represents genera of bacteria with little representation in the bacterial microenvironment.

## Data Availability

The microbiome database is available and can be downloaded at: http://www.ncbi.nlm.nih.gov/bioproject/807457 (accessed on 12 November 2024).

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
