# Peer review of "Changes in the Composition and Diversity of the Intestinal Microbiota Associated with Carbohydrate Consumption in Type 2 Diabetes Mellitus Patients"

_ijms, 2024, doi:10.3390/ijms252212359_

Round 1

Reviewer 1 Report

Comments and Suggestions for Authors

Major comments

The authors evaluated changes in the composition and diversity of the intestinal microbiota, associated with high carbohydrate consumption in type 2 diabetic and control patients. The authors found that patients that consumed high amounts of carbohydrates, regardless of whether they had diabetes mellitus and overweight, had a significantly reduced proportion of Faecalibacterium. The present study is interesting; however, some concerns should be addressed.

The introduction section is too short. It should review more adequately the pertinent literature notably concerning the different intestinal bacteria and their effects on health and the development of type 2 diabetes.

There are many spelling and punctuation errors in the text. Authors should revise the entire text.

In results section, page 2, the values of the HbA1c expressed in mg / dL should be checked. Moreover, the expression units of different parameters in table 1 should be presented.

Minor comments

Abstract

Some abbreviations have to be determined before using them in the text such as "FCF".

Materials and Methods

In lines 310-311, "One hundred subjects with and without a diagnosis of T2DM were invited to attend the hospital's outpatient clinic an informative talk" should be corrected as following: "One hundred subjects with and without a diagnosis of T2DM were invited to attend the hospital's outpatient clinic for an informative talk".

Results

In page 2, line 86, the word "fibber" should be corrected. Moreover, it is better to replace it with "dietary fiber".

Author Response

Reviewer 1

The authors evaluated changes in gut microbiota composition and diversity associated with high carbohydrate consumption in type 2 diabetic patients and control patients. The authors found that patients who consumed high amounts of carbohydrates, regardless of whether they had diabetes mellitus and were overweight, had a significantly reduced proportion of Faecalibacterium. The present study is interesting; however, some concerns should be addressed.

  1. The introduction is too short. The relevant literature should be reviewed more appropriately concerning the different intestinal bacteria and their effects on health and the development of type 2 diabetes.

ANSWERE:

This text was included in the introduction:

Evaluating the nutritional effects of carbohydrates is complex because food content is reported in different ways, for example, starch intake is often reported as total starch intake without specifying the subtype and digestibility [16]. Another factor that affects is that different terms can be used to describe the same carbohydrate, for example, when sugars are mentioned, they can refer to: sugars, total sugars, total available sugars, free sugars, added sugars, refined sugars, simple sugars or caloric sweeteners, this makes it impossible to assign a specific effect to a specific HCO, generating a confusing nutritional interpretation [17]. Therefore, it is difficult to evaluate and associate the consumption of a specific type of HCO with its nutritional impact on the microbiota and metabolism of individuals [18]. However, there are many studies that evaluate the different effects of T2DM on the body [19, 20, 21]. However, studies that evaluate the effect of carbohydrate consumption on the body and associated with T2DM are very scarce. There are some in animal models that show significant changes in the peripheral immune system [22, 23], of mucous membranes in the small intestine [24, 25], but in humans this relationship has been little evaluated.

  1. The text contains numerous spelling and punctuation errors. The authors should review the entire text.

ANSWER:

It was sent for proofreading; the certificate is attached.

  1. In the results section, page 2, the HbA1c values ​​expressed in mg/dL should be checked. In addition, the units of expression of the different parameters in Table 1 should be presented.

ANSWERE:

Numerical data are in table 1, therefore they were removed from the text and reference was made to Table 1.

The biochemical parameters of glycaemia and HbA1c, were found to be significantly elevated (P < 0.001) in the T2DM group, compared to the control group like shown in table 1.

Minor Comments

Abstract

  1. Some abbreviations need to be determined before use in the text, such as "FCF."

ANSWERE:

The abbreviations in the abstract have been revised, the following have been added:

Frequency Consumption Foods questionnaire (FCF) and Body Mass Index (BMI)

Materials and methods

  1. In lines 310-311, "One hundred subjects with and without a diagnosis of type 2 diabetes were invited to attend an informative talk in the hospital outpatient clinic" should be corrected as follows:

ANSWER: The paragraph was corrected in the text on page 10, lines 322-323."One hundred subjects with and without a diagnosis of type 2 diabetes mellitus were invited to attend an information talk at the hospital's outpatient clinic".

Results

  1. On page 2, line 86, the word "fiber" should be corrected. In addition, it is better to replace it with "dietary fiber."

ANSWER: The word "fiber" was replaced with "dietary fiber" on page 3, line 98.

Reviewer 2 Report

Comments and Suggestions for Authors

The manuscript titled "Changes in Composition and Diversity of the Intestinal Microbiota, Associated with Carbohydrate Consumption in Type 2 Diabetes Mellitus Patients" is an interesting, highly relevant, and up-to-date study. With some rigorous improvements, it can become an important resource for those in the medical field. Therefore:

The relationship between Bacteroides and T2DM is already known, as is the role of the Firmicutes/Bacteroides ratio. However, the purpose of the study is too general. Please elaborate and provide a more specific objective.

For the results section, the simultaneous presentation of statistical data can be replaced with a description of the findings, referring to the data in the tables.

For Figure 2, please add a general title before the descriptions of Figures 2A and 2B.

In the Materials and Methods section, please rename the subtitles to make the section clearer and easier to follow.

What types of carbohydrates were administered to the patients? Were they foods with added sugars or natural carbohydrates?

Although I understand that the study focuses on patients with T2DM, it would have been important to track the HOMA index to present the data correctly, especially considering that insulin resistance can significantly affect carbohydrate metabolism in diabetes. Please include this point in the Limitations section.

Finally, please make the Conclusions more concise, focusing solely on the main takeaways without repeating the results.

Please use the required settings, or use templates.

Author Response

Reviewer 2

The manuscript entitled "Changes in gut microbiota composition and diversity associated with carbohydrate consumption in patients with type 2 diabetes mellitus" is an interesting, highly relevant and up-to-date study. With some rigorous improvements, it can become an important resource for those working in the medical field. Therefore: The relationship between Bacteroides and T2DM is already known, as is the role of the Firmicutes/Bacteroides ratio.

  1. However, the purpose of the study is too general. Please explain and provide a more specific objective.

ANSWERE: In the introduction, a paragraph is expanded where more specific information about the objective is provided.

For the results section, the simultaneous presentation of statistical data can be replaced with a description of the findings, referring to the data in the tables.

ANSWERE:

The results were reviewed, and the data presented in Table 1 were removed from the text. Only the data that were not presented in any table, such as sociodemographic data, remained in the text (Page 2, lines 71-83).

The biochemical parameters of glycaemia and HbA1c, were found to be significantly elevated (P < 0.001) in the T2DM group, compared to the control group like shown in table 1. Page 2, line 85.

  1. For Figure 2, add a general title before the descriptions of Figures 2A and 2B.

ANSWERE:

If there is a general title that describes Figure 2, it is on page 2, lines 209-212, as shown below:

Figure 2a and 2b.

Structure bacterial communities of the feces samples of the CL and T2DM groups and their respective subgroups according to BMI. Average relative abundance at the Phylum level of the BMI subgroups. Only taxa with mean relative abundances > 0.05% were graphed. Control Normal Weight (CL-NW), Control Overweight (CL-OW), Type 2 Diabetes Mellitus Normal Weight (T2DM-NW), Type 2 Diabetes Mellitus Overweight (T2DM-OW).

  1. In the Materials and Methods section, rename the subheadings to make the section clearer and easier to follow.

ANSWERE:

The subtitles in the results section have been modified as shown below:

Sociodemographic data obtained from questionnaires applied to patients (page 2, line 70).

Anthropometric evaluation of patients with and without T2DM (page 2, line 80).

Biochemical evaluation of glycemia, HbA1c and triglycerides of patients with and without T2DM (page 2, line 84).

Dietary assessment from the Food Frequency Questionnaire (page 3, line 94).

Microbiota analysis of patients with and without T2DM (page 3, line 102).

  1. ¿Qué tipos de carbohidratos se administraron a los pacientes?, ¿Eran alimentos con azúcares añadidos o carbohidratos naturales?

ANSWERE: As described in the material and methods section, in this study, no carbohydrate was administered exogenously to the patients. An assessment of the dietary consumption of each patient was carried out using a 24-h food frequency questionnaire. From this questionnaire, the amount and type of carbohydrate consumed was determined (“Dietary evaluation and patient categorization”, page 10, line 336).

  1. While I understand that the study focuses on patients with type 2 diabetes, it would have been important to follow up on the HOMA index to present the data correctly, especially considering that insulin resistance can significantly affect carbohydrate metabolism in diabetes. Please include this point in the Limitations section.

ANSWERE:

It is not possible to include this point in the limitations section or to monitor the HOMA index because the objective of the study is not to monitor the metabolic effect of carbohydrates in diabetic patients. The main objective was: “to evaluate the changes in the composition and diversity of the intestinal microbiota associated with carbohydrate consumption in type 2 diabetes mellitus patients”. The methodology proposes a prospective, cross-sectional and comparative study design (page 10, line 311) with the determination of a single sample of biochemical parameters in addition to the collection of feces.

  1. Finally, make the Conclusions more concise, focusing only on the main findings without repeating the results.

ANSWERE:

The conclusions were worked on, made more concise, without repeating the results (page 12, lines 413-417).

  1. Use the required configuration or use templates.

ANSWERE:

The document is prepared with the required configuration and in the template proposed by the journal. If another template exists, it will be requested from the journal.

Reviewer 3 Report

Comments and Suggestions for Authors

Comments

1-     I am asking if you could add a dynamic group with possible antibiotics against  Bacteroidetes 244 54% and Firmicutes 45.23% . we do not know cause effect of the relation

2-     I am asking if you select possible probiotic therapy to improve DM parameters

3-     Please make sure that the references are consistent

10. Bridgewater LC, Zhang C, Wu Y, Hu W, Zhang Q, Wang J, et.al. Gender-based differences in host behavior and 466 gut microbiota composition in response to high fat diet and stress in a mouse model. Sci Rep 2017; 7(1):10776. 467

11. Rinninella E, Cintoni M, Raoul P, Lopetuso LR, Scaldaferri F, Pulcini G, et.al. Food Components and Dietary 468 Habits: Keys for a Healthy Gut Microbiota Composition. Nutrients 2019; 11(10):2393. 

I am asking how many names are listed in each reference before adding et al,

Author Response

Reviewer 3

  1. I am asking if you could add a dynamic group with possible antibiotics against Bacteroidetes 244 54% and Firmicutes 45.23%. We do not know the cause-effect relationship.

ANSWER:

For this document it is not possible to add it because the patients would not be the same. However, the working group is working on in vivo and in vitro studies with these parameters.

  1. I am asking if you select a possible probiotic therapy to improve DM parameters.

ANSWER:

It is possible for further research to select probiotic therapies or, where appropriate, explore those used by patients with T2DM to improve parameters.

  1. Make sure the references are consistent

ANSWERE:

All bibliographical references were reviewed, particularly these:

  1. Bridgewater, L.C., Zhang, C., Wu, Y. et al.Gender-based differences in host behavior and gut microbiota composition in response to high fat diet and stress in a mouse model. Sci Rep7, 10776 (2017). https://doi.org/10.1038/s41598-017-11069-4”
  2. Rinninella E, Cintoni M, Raoul P, Lopetuso LR, Scaldaferri F, Pulcini G, Miggiano GAD, Gasbarrini A, Mele MC. Food Components and Dietary Habits: Keys for a Healthy Gut Microbiota Composition. Nutrients. 2019; 11(10):2393.
  3. I'm asking how many names are listed in each reference before adding et al.

ANSWERE:

Citations were taken from PubMed's citation references and, where appropriate, the citation proposed by each journal. “The criteria used were those of IJ Molec Sci: reference list should include the full title, as recommended by the ACS style guide. Style files for Endnote and Zotero are available."

Round 2

Reviewer 1 Report

Comments and Suggestions for Authors

Major comments

Concerning the introduction, the authors did not respond to my previous comment. The authors should present the updated literature concerning the different intestinal bacteria and their effects on health and the development of type 2 diabetes.

In table 1 of the revised manuscript, the expression units of different parameters (HbA1c, TG, TC) are still missed. Moreover, the values of the HbA1c should be expressed as percentage.

Author Response

Reviewer 1

  1. Regarding the introduction, the authors did not respond to my previous comment. The authors should present updated literature on different intestinal bacteria and their effects on health and the development of type 2 diabetes.

ANSWERE:

The introduction was reviewed and adapted, and the changes made were highlighted in yellow:

Type 2 diabetes mellitus (T2DM) occurs when the pancreas does not produce enough insulin or the body does not use it effectively, which causes glucose intolerance [1]. In 2015, there was an estimated 415 million people with DM; by 2040, the number will be 642 million [2]. In Mexico, T2DM is the main cause of morbidity and mortality among individuals aged between 45 and 64 years [3]. It is a multifactorial disease involving genetic, dietary, lifestyle (e.g., sedentarism) [4], and environmental factors (e.g., intestinal microbiota) [5]. Intestinal microbial composition is considered a factor of environmental origin because it differs among patients according to their environments [6]. Recent studies propose a link between the composition of the intestinal microbiota, its metabolites and the health of individuals [7, 8]. Accordingly, the type, quantity, and quality of the diet modifies patients’ microbiota, which causes dysbiosis that influences the pathophysiology of T2DM [9], however, it is still unclear how these mechanisms are articulated. The intestinal microbiota is defined as a complex and diverse community dominated by bacteria living in the intestine [10]. In healthy adults, the most abundant bacterial phyla are Bacteroidetes (Gram-negative bacteria) and Firmicutes (Gram-positive bacteria) [11]; Actinobacteria, Fusobacterium, and Verrucomicrobia are variable among individuals [12], but this composition can variety in some pathologies. Several studies show that the type of microbes present in the intestine drive and influence some of the metabolic modifications in individuals that cause the presence of obesity and T2DM [13-18]. A high sugar diet is associated with symptoms of type 2 diabetes in rats, with inflammation, alterations in glucose metabolism and intestinal greater abundance presence of harmful bacteria [19]. In addition to, it has been reported that modifying the composition of the intestinal microbiota improves insulin sensitivity [20] and that the low abundance of Lactobacillus, Prevotella, Bacteroides, Desulfovibrio and Oxalobacter spp promotes metabolic disorders such as Insulin Resistance (IR) [21]. However, in patients with T2DM, but thin, a reduced abundance of Akkermansia muciniphila has been shown, also associated with a reduction in insulin secretion [22-24]. Other microorganisms related to metabolic effects in T2DM are Ruminococcus, Clostridium, Bifidobacterium, Bacteroides, Eubacterium, Listeria among others [25, 26]. The presence of Blastocystis spp and Prevotella copri are indicators of a favorable postprandial glucose metabolism [27], therefore, the alteration of their abundance generates a greater risk of developing T2DM with alteration in postprandial glucose metabolism [28].

Each nutrient consumed in the diet alters the composition of the intestinal microbiota, promoting the expansion of microorganisms [11]. Studies have described the effects of high-fat diets [12], but there have been no studies on the impact of consuming different types of carbohydrates (CHO) on the intestinal microbiota. For example, most individuals consume different proportions of sucrose as a dietary additive [13]. Hydrolyzed sucrose is retained in the distal intestine, exposing the microbiota to significant amounts of fructose and glucose for a longer period [14]. Sucrose has been found to decrease the abundance of Bacteroides thetaiotaomicron, which are important regulators of intestinal colonization [15]. Evaluating the nutritional effects of CHO is complex because food content is reported in different ways; for example, starch intake is often reported as total starch intake without specifying the subtype and digestibility [16]. Another factor is that different terms can be used to describe the same CHO; for example, sugars can refer to sugars, total sugars, total available sugars, free sugars, added sugars, refined sugars, simple sugars, or caloric sweeteners. This makes it impossible to assign a specific effect to a specific CHO, leading to a confusing nutritional interpretation [17]. Therefore, it is difficult to evaluate and associate the consumption of a specific type of CHO with its nutritional impact on the microbiota and metabolism of individuals [18, 22]. Although several studies have evaluated the effects of T2DM on the body [19, 20, 21), studies evaluating the effects of CHO consumption on the body and associated with T2DM are scarce. Some animal models have shown significant changes in the peripheral immune system [22, 23] and mucous membranes in the small intestine [24, 25], but there has been little evaluation of this relationship in humans.

Thus, the objective of this study was to evaluate the changes in composition and diversity of the intestinal microbiota associated with CHO consumption in T2DM patients.

  1. In Table 1 of the revised manuscript, the units of expression for the different parameters (HbA1c %, TG mg/dL, TC mg/dL) are still missing. In addition, HbA1c values ​​should be expressed as a percentage.

ANSWER:

The units of expression for the biochemical parameters were placed in yellow, table 1 page 4, line 154.

Reviewer 2 Report

Comments and Suggestions for Authors

Thank you for the opportunity to review this article. Thanks for correcting each point, thus I recommend for publication. Congratulations on completing the manuscript!

Author Response

Thank you, there are no comments to reply to.

Round 3

Reviewer 1 Report

Comments and Suggestions for Authors

The authors responded to my previous commentaries.